# THINKBOT: EMBODIED INSTRUCTION FOLLOWING WITH THOUGHT CHAIN REASONING

**Guanxing Lu[1], Ziwei Wang[2♯], Changliu Liu[3], Jiwen Lu[4], Yansong Tang[1†]**

[1]Tsinghua Shenzhen International Graduate School, Tsinghua University
[2]School of Electrical and Electronic Engineering, Nanyang Technological University
[3]Carnegie Mellon University   [4]Department of Automation, Tsinghua University
{lgx23@mails.,lujiwen@,tang.yansong@sz.}tsinghua.edu.cn
ziwei.wang@ntu.edu.sg  cliu6@andrew.cmu.edu

## ABSTRACT

Embodied Instruction Following (EIF) requires agents to complete human instruction by interacting objects in complicated surrounding environments. Conventional methods directly consider the sparse human instruction to generate action plans for agents, which usually fail to achieve human goals because of the instruction incoherence in action descriptions. On the contrary, we propose ThinkBot that reasons the thought chain in human instruction to recover the missing action descriptions, so that the agent can successfully complete human goals by following the coherent instruction. Specifically, we first design an instruction completer based on large language models to recover the missing actions with interacted objects between consecutive human instruction, where the perceived surrounding environments and the completed sub-goals are considered for instruction completion. Based on the partially observed scene semantic maps, we present an object localizer to infer the position of interacted objects and the related Bayesian uncertainty for close-loop planning. Extensive experiments in the simulated environment show that our ThinkBot outperforms the state-of-the-art EIF methods by a sizable margin in both success rate and execution efficiency. Project page: https://guanxinglu.github.io/thinkbot/.

## 1 INTRODUCTION

Designing autonomous agents for diverse household tasks has been highly desired in research of artificial intelligence for a long time. Recent advances in computer vision Wang et al. (2023a); Li et al. (2023b) and natural language processing Brown et al. (2020); Ouyang et al. (2022) enable autonomous agents to complete complex human requirements, because the appeared large pre-trained models can comprehend human instruction and perceive the world accurately. Embodied instruction following (EIF) Misra et al. (2017); Zhu et al. (2017); Gordon et al. (2018); Shridhar et al. (2020) requires the agent to ground human instruction to consecutive task plans with feasible execution, which necessitates high success rate and completion efficiency.

To accomplish the challenging EIF task, imitation learning Shridhar et al. (2020); Pashevich et al. (2021); Singh et al. (2021); Song et al. (2022); Nguyen et al. (2021); Zhang & Chai (2021); Nguyen & Okatani (2020); Suglia et al. (2021); Kim et al. (2021) is widely adopted to generate low-level actions from historical observation and given instruction, while they fail to adapt to new scenarios due to the insufficient pair-wise data between human instruction and low-level actions. To this end, modular methods Blukis et al. (2022a); Min et al. (2022); Murray & Cakmak (2022); Inoue & Ohashi (2022); Kim et al. (2023); Ding et al. (2022); Liu et al. (2022) decompose complex tasks to high-level planning conditioned on instruction and low-level execution with predefined controllers. However, human instruction is usually sparse with incoherence for action plan generation of agents. For example, in Figure 1, humans may give the instruction *'Prepare a spoon, Take a mug'* for

---

♯ Project lead. † Corresponding author.

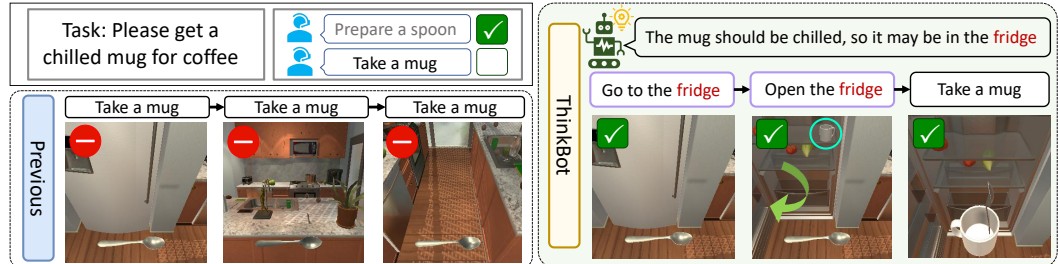

Figure 1: Comparison between conventional EIF methods (Prompter Inoue & Ohashi (2022)) and our ThinkBot. Existing methods directly leverage sparse human instruction to generate action sequence, which usually get stuck due to the incoherence of instruction. Our ThinkBot recovers missing action descriptions by reasoning the thought chain in sparse human instruction, and can successfully complete challenging tasks.

making cold brew coffee. In realistic scenarios, the mug may be stored in a fridge, and the agent may need to open the fridge first to get the mug. Therefore, the dense instruction should be *'Go to the fridge, open the fridge, and take a mug'*. Lacking coherent instruction usually disables the agent to acquire feasible action sequences in execution, and the success rate across diverse tasks in complicated indoor environments still remains low.

In this paper, we propose a ThinkBot agent to accurately complete diverse EIF tasks in interactive environments. Unlike existing methods that directly employ the sparse human instruction for agent action sequence generation, we recover the missing action descriptions for agent execution by reasoning the thought chain in sparse human instruction. Therefore, coherent instruction can be leveraged to generate more feasible agent actions in complex EIF tasks especially with long sequences. More specifically, we first propose an instruction completer based on large language models to predict the missing actions with interacted objects in the sparse human instruction, where we carefully design prompts to consider the perceived objects in the surrounding environments and the completed sub-goals in the task execution sequences. We then present a multimodal object localizer that predicts the position of the recovered missing objects based on the scene semantic maps, where the mined object correlation is also leveraged to enhance the localization ability. Extensive experiments on ALFRED Shridhar et al. (2020) show that our ThinkBot outperforms the state-of-the-art EIF methods by a sizable margin in both success rate and execution efficiency. Our main contributions can be summarized as follows:

- We propose a ThinkBot agent that reasons the thought chain in sparse human instruction for coherence mining to successfully complete complex EIF goals.
- We present an instruction completer based on large language models to generate the missing actions with interacted objects, and propose an object localizer to predict the position of objects for interaction.
- We conduct extensive experiments of diverse EIF tasks on ALFRED benchmark, and the results demonstrate that our method achieves higher success rate and path-length-weighted success rate than the state-of-the-art methods on unseen environments.

## 2 RELATED WORK

**Embodied Instruction Following:** Developing generalist agents that can follow human instruction to complete diverse tasks in interactive environments is a long-standing goal. In the pursuit of this goal, EIF has been widely studied in recent years for high generalizability and practicality. Prior arts can be divided into two categories: end-to-end methods and modular methods. End-to-end methods directly generate low-level actions conditioned on the current state of the environment and human instruction Shridhar et al. (2020); Pashevich et al. (2021); Singh et al. (2021); Song et al. (2022); Nguyen et al. (2021); Zhang & Chai (2021); Nguyen & Okatani (2020); Suglia et al. (2021); Kim et al. (2021); Yang et al. (2018). For instance, Pashevich et al. Pashevich et al. (2021) developed an episodic transformer to encode language inputs and the episode history of visual observation and actions, which was decoded for action sequence generation with auto-regression. However, end-to-

end methods often struggle to generalize to unseen scenes due to insufficient pair-wise data between instruction and low-level action sequences. To address this, modular methods Jia et al. (2022); Blukis et al. (2022a); Min et al. (2022); Murray & Cakmak (2022); Inoue & Ohashi (2022); Kim et al. (2023); Ding et al. (2022); Liu et al. (2022) plan high-level action sequences and execute them with pre-defined local policies guided by online semantic maps, which are free of the pair-wise data between instruction and low-level actions. In modular methods, selecting correct targets and actions for navigation and interaction is important for searching efficiency and task success. Min et al. Min et al. (2022) directly employed convolutional networks to predict the target position from current semantic map, and Murray et al. Murray & Cakmak (2022) proposed a landmark classification model based on BERT Kenton & Toutanova (2019) for target selection from original human instruction. Inoue et al. Inoue & Ohashi (2022) predicted landmark objects according to offline co-occurrence probabilities of objects evaluated by pre-trained language models, and Kim et al. Kim et al. (2023) yield the detailed plan by incorporating the contextual information of natural language instructions. Nevertheless, directly considering the sparse human instruction for agent action generation usually fails to achieve human goals due to the instruction incoherence with missing action descriptions.

**LLMs for Embodied Agents:** Large language models (LLMs) Brown et al. (2020); Ouyang et al. (2022) have demonstrated their capability in embodied AI tasks such as visual-language navigation Qiao et al. (2023); Long et al. (2023); Shah et al. (2023); Zhou et al. (2023a); Georgakis et al. (2022); Chen et al. (2022), object navigation Yu et al. (2023); Zhou et al. (2023b), open-world exploration Wang et al. (2023b); Zhu et al. (2023); Chen et al. (2024), and embodied planning Mu et al. (2023); Yao et al. (2022); Wu et al. (2023); Ahn et al. (2022); Huang et al. (2022a;b); Raman et al. (2022); Singh et al. (2023); Lu et al. (2022), where the high generalizability across deployment scenes and downstream tasks of LLMs enables embodied agents to achieve diverse human goals in complex environments. With the rich commonsense embedded in LLMs, fine-grained actions regarding human instruction can be directly generated. Zhou et al. Zhou et al. (2023a) directly prompted LLMs to perform zero-shot sequential action prediction by taking the textual descriptions of historical visual observations as inputs. To facilitate efficient exploration, ESC Zhou et al. (2023b) generated frontier candidates on the observed semantic map, and employed LLM to determine the next frontier by considering hand-crafted soft constraints jointly. While direct generation of fine-grained actions is challenging for LLMs due to the extremely large search space, other methods decompose the overall solution into high-level plan generation and low-level action controlling Mu et al. (2023); Yao et al. (2022); Wu et al. (2023); Ahn et al. (2022); Huang et al. (2022a;b). Wu et al. Wu et al. (2023) crafted a large-scale embodied planning dataset and finetuned different plan generators on the dataset for task plan grounding. Yao et al. Yao et al. (2022) generated consecutive plans by prompting LLMs to synergize reasoning and acting, and achieved impressive performance on text-based benchmarks Shridhar et al. (2021). However, despite the high reasoning ability of LLMs, the spatial localization ability for interacted objects is weak in LLMs. We present a multimodal transformer-based object localizer to provide spatial guidance and feedback for agents in object interaction accurately.

## 3 APPROACH

In this section, we first briefly introduce the problem in EIF and our overall pipeline of the ThinkBot agent (Section 3.1). Then we detail the instruction completer that recovers the missing actions with interacted objects for coherence mining (Section 3.2), and demonstrate the multimodal object localizer which provides spatial guidance for agents in interaction (Section 3.3).

### 3.1 PROBLEM STATEMENT AND OVERALL PIPELINE

The embodied instruction following task requires an agent in the interactive environment to finish the human goals physically by generating action sequences, where the human goals and the step-by-step instruction are given to the agent for guidance. Following the modular methods, we first generate high-level subgoal sequences and then execute them with a pre-defined controller guided by online semantic maps. In the $t_{th}$ step, the agent needs to generate a high-level subgoal $A_t$ based on the current instruction $I_t$ and the object state $S_{t-1}$ after implementing the last subgoal by the controller. A high-level plan $A_t$ is a tuple $(a_t, o_t, p_t)$, where $a_t$ is the primitive action and $o_t$ means the interacted object with the position $p_t$. In EIF tasks, human instruction is usually

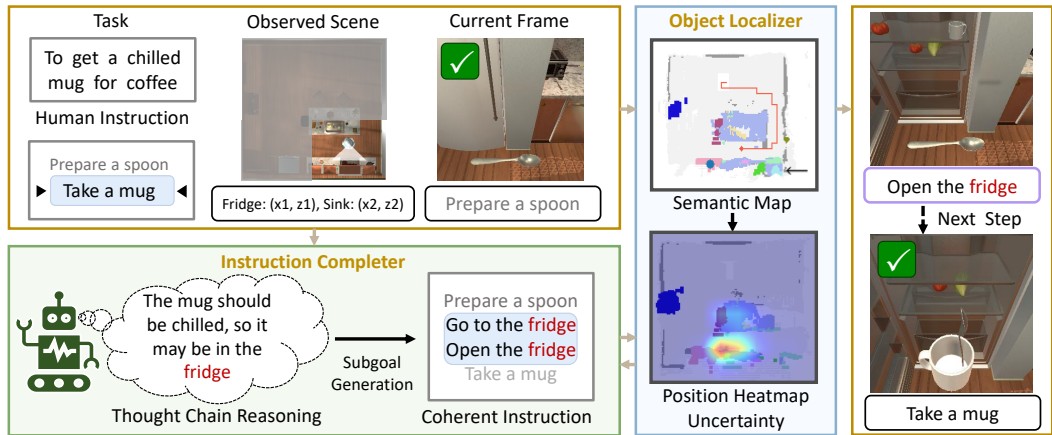

Figure 2: The overall pipeline of ThinkBot, which consists of an instruction completer and an object localizer. The instruction completer generates the coherent instruction with interacted objects based on sparse human instruction and the current visual perception results, and the object localizer predicts the position of the interacted object for manipulation and navigation.

sparse with significant incoherence between consecutive steps. Therefore, it is very challenging to generate feasible action $A_t$ based on the current step-wise instruction $I_t$ and the object state $S_{t-1}$ after implementing the last action. For example, humans may provide the instruction *'Prepare a spoon, take a mug'* for coffee making. However, the mug may be stored in the fridge in realistic scenes. Consequently, it is difficult for the agent to take the mug with spoon without other instruction after preparing the bread. The coherent instruction should be *'Prepare a spoon, go to the fridge, open the fridge, take a mug'*. Therefore, our goal is to recover the missing action descriptions in the sparse human instruction.

Since the thought chain reveals a series of intermediate steps from the initial problem to the final solution Wei et al. (2022), it can be leveraged to decompose the original complex problems and enhance the feasibility of the solution. In EIF tasks, reasoning the thought chain can predict the missing action descriptions in the sparse human instruction to successfully achieve the goal. The overall pipeline of our ThinkBot agent is shown in Figure 2, which consists of an instruction completer recovering the missing actions with interacted objects and an object localizer predicting the object location for agent interaction. For the instruction completer, we design prompts for the pre-trained large language model including the descriptions of scene information, task completion process, and the executor feedback which is expected to reason the thought chain in sparse human instruction to provide coherent instruction. For the object localizer, the generated missing actions with interacted objects and the perceived semantic map of the scene are utilized to predict the object location for the agent to interact with, where multimodal transformers are employed for the alignment between language instruction and visual clues in the environment. Finally, the agent can easily complete the human goals to achieve significantly higher success rate with the coherent instruction and the explicit interaction location.

## 3.2 INSTRUCTION COMPLETER

To recover the missing action descriptions in the sparse human instruction, we employ LLMs with rich commonsense to reason the thought chain in the instruction. While LLMs have demonstrated remarkable abilities in various tasks, unlocking their complete potential requires prompt engineering. To enable the pre-trained LLMs to predict the missing actions with interacted objects accurately, we carefully design the prompt by organizing the system message and agent message that respectively describe the innate world properties and the perceived information. The input and output of the instruction completer are shown in Figure 3 with an example.

**System Message:** The system message describes the innate world properties in the simulator including the role explanation, message definition, primitive actions and response format, which remains unchanged during the whole EIF process for the given task and environment. The role explanation

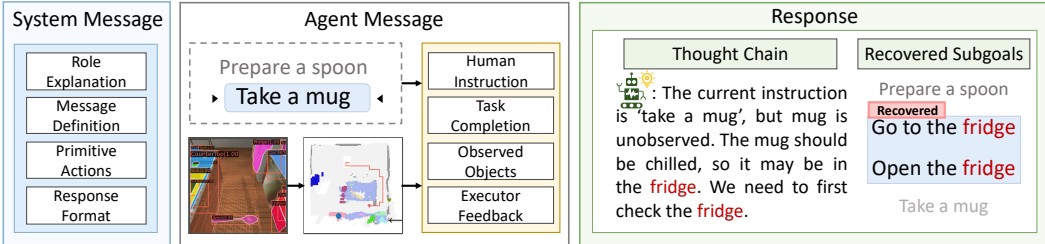

Figure 3: Input and output of the instruction completer based on LLMs. The input contains system message describing the world properties and agent message demonstrating perceived environment information. The output includes the thought chain in sparse human instruction and missing subgoals with interacted objects.

defines the household AI assistant's scenario, message definition explains input meanings, primitive actions limit agent options, and response format structures LLM outputs for missing actions.

**Agent Message:** The agent message demonstrates the perceived information including the human instruction, task completion process, observed objects and the executor feedback, which is updated along with the EIF process. The human instruction means the given sparse instruction sentences and the final goal. The task completion depicts completed subgoals, the current subgoal and all subgoals, which reflects the execution process of the final goal. The information on observed objects in the scene is represented by the object category and the coordination of the instances that have been seen by the agent, which demonstrates the affordable candidates for interaction during the EIF process. The executor feedback contains the uncertainty obtained by the multi-modal object localizer and the environmental feedback to enable close-loop planning. Finally, the system message and the agent message are concatenated to form the prompt for recovering missing actions with interacted objects in the sparse instruction.

**Uncertainty-aware Thought Chain Reasoning:** To accurately recover the missing actions in sparse human instruction, we enforce LLMs to reason the thought chain in human instruction that indicates the detailed process from the initial state to the final goal. However, the recovered actions can only be executed if the agent knows the exact position of the interacted objects. Therefore, we also require LLMs to generate the missing subgoals in a structured format for subsequent location prediction. The recovered subgoals are leveraged in subsequent object localizer for position prediction in the partially-observed map, where the uncertainty of the object location is also outputted. Then, the LLMs leverage the predicted uncertainty provided in the executor feedback and the object location to refine the subgoals, so that the agent can manipulate the object to successfully achieve the subgoal for task completion.

## 3.3 MULTIMODAL OBJECT LOCALIZER

LLMs can reason through sparse instructions to recover missing actions with objects, but struggle with spatial localization due to weak language-based positioning. To address this, we propose a multimodal transformer-based object localizer in Figure 4 that predicts object positions and related Bayesian uncertainty using recovered instructions and a partially observed semantic map.

To obtain the semantic map, the egocentric image is used to estimate a depth map. At the same time, an instance segmentation mask is created with a pretrained detector. These two results are combined to create a semantic 3D point cloud, which is then changed into the partially observed 2D semantic map. For the instruction encoder, we leverage a pretrained BERT Kenton & Toutanova (2019) to extract the instruction features $\mathbf{X}_s$ for the human instruction and the goal object, which are acquired from the prediction results of instruction completer. For the map encoder, we employ convolutional neural networks to extract the initial map features $\mathbf{X}'_t$. Since semantic cor-

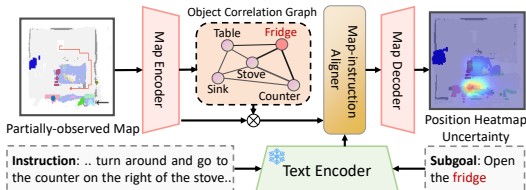

Figure 4: The architecture of the multimodal object localizer, where the object correlation graph is also learned to strengthen the map features.

relation among objects provides beneficial priors for accurate localization, we mine the object correlation graph to embed the priors into semantic map features for further enhancement of object position prediction. The graph is defined as $\mathcal{G} = (\mathbf{V}, \mathbf{E})$, where $\mathbf{V}$ is the set of nodes and $\mathbf{E}$ are the edges between different nodes. Each node represents a possible object category in the simulator, and the element in the $i_{th}$ row and $j_{th}$ column of $\mathbf{E}$ indicates the correlation of objects between the $i_{th}$ and the $j_{th}$ classes. While conventional graph convolutional networks use external priors to handcraft a predefined graph structure, we construct the object correlation graph by learning from the semantic map features $\mathbf{X}'_t$ to adapt to different scenes. The object correlation graph is generated from the semantic map features $\mathbf{X}'_t$ by $\mathbf{E}_t = f(\mathbf{X}'_t \mathbf{W}_e)$, where $f(\cdot)$ denotes the activation function and $\mathbf{W}_e$ is a learnable matrix for graph generation. The process of learning the object correlation graph can be regarded as encoding the object correlation priors among the categories. We then employ graph convolutional layers to enhance the semantic map representation with the object correlation encoding, and acquire the map features for multimodal alignment according to the following:

$$\mathbf{X}_t = \mathbf{X}'_t + \mathbf{E}\mathbf{X}'_t\mathbf{W}_a, \tag{1}$$

where $\mathbf{W}_a$ is a learnable weight matrix for message passing to embed the object correlation priors into the map features. After the representation extraction for the instruction and the semantic map, we leverage a map-instruction aligner to align the instruction features and the map features for object position prediction. Since the semantic map is updated in an online manner with high frequency during agent navigation, we take the features of the semantic maps as the query. The instruction remains unchanged in a subtask, which is utilized as the key and the value features. The representation that aligns the visual information provided by semantic maps and the language information in the instruction is acquired by cross-attention, which is then leveraged to decode the predicted location of the interacted objects.

In addition to predicting the object location, we also allow the network to output an uncertainty measured by the variance, which can serve as feedback for our instruction completer. However, obtaining annotated uncertainty labels can be costly. To address this, we leverage the Bayesian uncertainty estimation to learn the aleatoric uncertainty Kendall & Gal (2017); Ding et al. (2022) to model the probabilistic nature of unseen object localization. Specifically, we assign a scalar variance to each pixel that measures the prediction uncertainty of this pixel, which encourages the instruction completer to refine the current plan from exploitation to exploration if the prediction has low confidence. To train the object localizer, we need to acquire the groundtruth instance position $y$ for the interacted objects that are predicted by the instruction completer. We replay expert demonstrations in the training data, and record the masks of interacted objects when a subtask is successfully completed. With the projection from the egocentric mask to the top-down view by depth estimation, the groundtruth positions are used to supervise the object localizer to predict the position of interacted objects $\hat{y}$. The labels of pixels covered by the interacted objects are set to one and otherwise to zero. The model is trained with a combination of pixel-wise binary cross-entropy loss and the uncertainty regularization term:

$$\mathcal{L} = \frac{1}{T} \sum y \log(\hat{y}_t) + (1 - y) \log(1 - \hat{y}_t), \tag{2}$$

where the output with uncertainty estimation is given by $\hat{y}_t = \hat{y} + \epsilon_t$ at sample $t$, and $\epsilon_t \sim N(0, \sigma^2)$ is a sampled Gaussian noise based on the estimated variance. $T$ denotes the total sampling time. By optimizing this objective, the training sample that hard to locate is adaptively assigned with a high variance, thus realize the uncertainty estimation that can be provided to the instruction completer as feedback signal, which shows effectiveness in both quantitative and qualitative analysis.

## 4 EXPERIMENTS

In this section, we first introduce the experiment setup including datasets, baseline methods, evaluation metrics and implementation details. Then we compare our method with the state-of-the-art EIF approaches to show the superiority in success rate and efficiency, and conduct an ablation study to verify the effectiveness of the instruction completer and the object localizer. Finally, we also demonstrate the visualization results of our method to depict our intuition. Additional results and case studies are provided in the appendix.

Table 1: Comparison with the state-of-the-art methods in SR, GC, PLWSR, PLWGC on the test seen and test unseen splits.

| Method | Test Seen | | | | Test Unseen | | | |
|---|---|---|---|---|---|---|---|---|
| | PLWGC | GC | PLWSR | SR | PLWGC | GC | PLWSR | SR |
| Seq2seq | 6.27 | 9.42 | 2.02 | 3.98 | 4.26 | 7.03 | 0.08 | 3.9 |
| MOCA | 22.05 | 28.29 | 15.10 | 22.05 | 9.99 | 14.28 | 2.72 | 5.30 |
| E.T. | 34.93 | 45.44 | 27.78 | 38.42 | 11.46 | 18.56 | 4.10 | 8.57 |
| LWIT | 23.10 | 40.53 | 43.10 | 30.92 | 16.34 | 20.91 | 5.60 | 9.42 |
| HITUT | 17.41 | 29.97 | 11.10 | 21.27 | 11.51 | 20.31 | 5.86 | 13.87 |
| ABP | 4.92 | 51.13 | 3.88 | 44.55 | 2.22 | 24.76 | 1.08 | 15.43 |
| LLM-Planner | - | 26.77 | - | 18.20 | - | 23.37 | - | 16.42 |
| FILM | 15.59 | 39.55 | 11.27 | 28.83 | 15.13 | 38.52 | 11.32 | 27.80 |
| LGS-RPA | 28.97 | 48.66 | 21.28 | 40.05 | 22.76 | 45.24 | 22.76 | 35.41 |
| Prompter | 30.72 | 63.43 | 25.81 | 53.23 | 26.22 | 58.76 | 20.76 | 45.72 |
| CPEM | 27.49 | 59.40 | 22.61 | 50.62 | 27.00 | 61.10 | 22.61 | 49.84 |
| Prompter+ | 36.35 | 70.20 | 31.12 | 60.86 | 30.09 | 65.71 | 26.22 | 55.46 |
| **ThinkBot** (Ours) | **37.01** | **71.64** | **32.02** | **62.69** | **30.73** | **67.75** | **26.93** | **57.82** |

## 4.1 EXPERIMENTAL SETUP

**Dataset:** For the simulation of EIF tasks, we utilize the well-recognized ALFRED benchmark Shridhar et al. (2020) within the AI2-THOR Kolve et al. (2017) virtual environment. The ALFRED benchmark includes 25,743 trajectory-instruction pairs, covering 7 different task types with varying levels of complexity. The benchmark is divided into five splits including *train*, *test seen*, *test unseen*, *valid seen* and *valid unseen*. The ALFRED benchmark poses significant challenges for EIF agents, as it requires them to ground incoherent natural instruction of different granularity into various household tasks that involve long-horizon reasoning plans. To further evaluate the generalizability and the planning accuracy of ThinkBot, we also evaluate it on ActioNet Duan et al. (2020), which contains a variety of challenging high-level planning tasks that differs from ALFRED.

**Baselines:** We compare our agent, ThinkBot, with previously published state-of-the-art EIF models. The counterparts include end-to-end methods Seq2seq Shridhar et al. (2020), MOCA Singh et al. (2021), E.T. Pashevich et al. (2021), LWIT Nguyen et al. (2021), HITUT Zhang & Chai (2021), ABP Kim et al. (2021), and modular methods LLM-Planner Song et al. (2023), FILM Min et al. (2022), LGS-RPA Murray & Cakmak (2022), Prompter Inoue & Ohashi (2022), CPEM Kim et al. (2023). We also construct a strong baseline denoted as Prompter+ in our experiments, which is a modified version of Prompter Inoue & Ohashi (2022) that combines environment-aware memory Kim et al. (2023) and a re-trained object detector Wang et al. (2023a).

**Evaluation Metrics:** We follow the evaluation protocol outlined in the ALFRED benchmark. The primary metric is the success rate (SR) that measures the percentage of tasks completed, and we also report the goal-condition success rate (GC), which evaluates the percentage of satisfied goal conditions for all subgoals in step-by-step instruction. To account for efficiency in task completion, both SR and GC are penalized by the length of the execution sequence to compute a path-length-weighted (PLW) score for each metric, which are termed PLWSR and PLWGC respectively. To evaluate the planning outputs of our ThinkBot, We report the high-level planning (HLP) accuracy from Song et al. (2023) compared with atomic actions for ALFRED, and report BERT similarity compared with dense instructions for ActioNet.

**Implementation Details:** The instruction completer adopts the publicly released GPT-3.5 API `GPT-3.5-turbo` as the base model, where we set the generation temperature to 0 for stability enhancement. For prompt design, we leverage emotion prompt Li et al. (2023a) and prompt optimization Yang et al. (2023) in the system message template to further boost the performance of LLMs. For the multimodal object localizer, we employ a truncated ResNet18 Georgakis et al. (2022) for the map encoder. AdamW optimizer Loshchilov & Hutter (2017) with the initial learning rate $5 \times 10^{-4}$ and step decay is employed for parameter update.

## 4.2 COMPARISON WITH THE STATE-OF-THE-ART METHODS

**Comparison on Task Completion:** We compare the proposed ThinkBot with the state-of-the-art methods on the ALFRED benchmark[1]. Table 1 illustrates the comparison of SR,

---

[1]The results have been publicly released in https://leaderboard.allenai.org/alfred/submissions/public

Table 2: Comparison of methods combining different proposed techniques, where valid unseen and the selected hard valid unseen splits are used for evaluation.

| Method | Valid Unseen | | | | Hard Valid Unseen | | | |
|---|---|---|---|---|---|---|---|---|
| | PLWGC | GC | PLWSR | SR | PLWGC | GC | PLWSR | SR |
| Random | 26.18 | 67.64 | 23.80 | 59.68 | 0.32 | 5.41 | 0 | 0 |
| FILM | 28.74 | 72.46 | 26.58 | 64.31 | 0.29 | 4.76 | 0 | 0 |
| Prompter+ | 29.36 | 72.00 | 26.82 | 64.43 | 0.48 | 5.41 | 0 | 0 |
| Groundtruth Location | 39.71 | 72.75 | 37.01 | 67.97 | 0.79 | 5.41 | 0 | 0 |
| w/o Instruction Completer | 29.09 | 72.38 | 26.43 | 64.92 | 0.48 | 5.41 | 0 | 0 |
| w/o Object Localizer | 30.24 | 74.37 | 27.87 | 66.99 | 9.29 | 22.41 | 8.11 | 16.22 |
| w/o Object Correlation Graph | 30.41 | 73.89 | 28.14 | 67.36 | 11.31 | 29.46 | 9.74 | 21.62 |
| **ThinkBot** | **31.11** | **75.30** | **28.73** | **67.72** | **11.95** | **30.86** | **10.26** | **22.97** |

GC, PLWGC, and PLWSR on the test splits for both seen and unseen scenarios. Our ThinkBot achieves the best performance on all four metrics in both test seen and test unseen split, outperforming the previous arts including both end-to-end and modular methods by a sizable margin. Compared with CPEM Kim et al. (2023), ThinkBot surpasses by 7.98% and 12.07% SR on the test unseen and test seen split, respectively.

Besides, Prompter leverages the large language models to infer object co-occurrence probability for semantic search in EIF, and is further enhanced with better visual perception modules to acquire our Prompter+. However, they still suffer from the sparse human instruction with incoherence that usually causes execution failure. On the contrary, our ThinkBot reasons the thought chain in the sparse human instruction to recover the missing action descriptions, which can provide coherent instruction for the agent to successfully

Table 3: Comparison with the state-of-the-art method in HLP accuracy and BERT similarity.

| Method/Dataset | ALFRED | ActioNet |
|---|---|---|
| LLM-Planner | 43.2 | 68.4 |
| **Thinkbot** | **69.9** | **73.3** |

complete the human goal with high efficiency. As a result, our method outperforms the second-best Prompter+ method by 1.83% (62.69% vs. 60.86%) and 1.44% (71.64% vs. 70.20%) in SR and GC respectively in the test seen split. Moreover, the advantages of our method are more obvious in the test unseen split, which are 2.36% (57.82% vs. 55.46%) in SR and 2.04% (67.75% vs. 65.71%) in GC. The results indicate the high generalization ability of our ThinkBot even in novel scenarios that are never seen in training data. In terms of the efficiency in task completion, our method achieves 32.02% PLWSR and 37.01% PLWGC in the test seen split, and 26.93% PLWSR and 30.73% PLWGC in the test unseen split, which demonstrates the efficiency of our method. In conclusion, our ThinkBot agent is more practical than stat-of-the-art methods for EIF tasks in scenarios with complex environments and long sequences.

**Comparison on High-level Planning Accuracy:** To assess the generalizability of Thinkbot, we evaluate the outputs of the instruction completer in both ALFRED and ActioNet in Table 3. The results show that our method outperforms Song et al. (2023) significantly in both ALFRED and ActioNet datasets, which indicates the generalizabilty of our ThinkBot provided by the visual grounding and feedback mechanism.

## 4.3 ABLATION STUDY

Our instruction completer recovers the missing actions with interacted objects by reasoning thought chain in sparse human instruction, and the object localizer predicts the position of the interacted objects. In Table 2, we conduct an ablation study to verify the effectiveness of each presented technique. Since recovering the incoherent instruction is especially beneficial in the hard cases where target objects are located inside closed containers (e.g. The task *'to get a chilled mug for coffee'* where the mug is in a closed fridge), we also evaluate different methods in the hard cases extracted from valid unseen split, referred to as *hard valid unseen*. In this subset, the agent must open the receptacles to locate the target objects by recovering the missing interaction with the receptacles from sparse instruction, and then the challenging tasks in the hard cases can be completed.

**Instruction Completer:** We first implement our ThinkBot without the instruction completer, where we directly predict the location of the target in the original sparse human instruction with the object localizer for interaction. The results in success rate drops 2.80% compared with our vanilla ThinkBot in the valid unseen split. Directly predicting the location of interacted objects in incoherent instruction usually causes large deviation, because the semantic correlation is weak between the target object and the observed semantic map. In hard cases, the agent is more likely to fail to

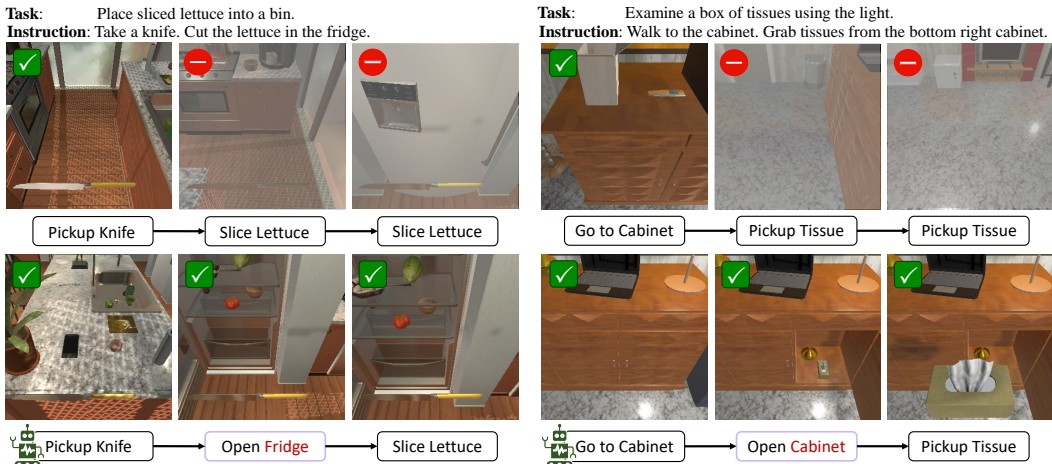

Figure 5: Visualization of the agent action sequence acquired by Prompter+ (top) and our ThinkBot (bottom), where our method can recover the missing actions with interacted instances 'Open Fridge' and 'Open Cabinet' to successfully achieve the human goal.

Figure 6: Additional visualization of the complete agent action sequence acquired by our ThinkBot, where our method can not only recover the missing actions with interacted instances but also revise the recovered actions when opening the wrong cabinet.

complete the task due to the missing interaction with the containers, and the invisible target object is hard to discover by the agent. As a result, the performance in SR without instruction completer drops to zero in hard unseen split. On the contrary, our instruction completer can reason intermediate action descriptions to mine the semantic correlation, which provides fine-grained instruction for the agent to achieve human goals.

**Object Localizer:** We also evaluate the performance of our ThinkBot without the object localizer, and observe notable performance drops in SR and PLWSR. In the valid unseen split, the performance drops 0.73% in SR and 0.86% in PLWSR compared with our vanilla ThinkBot. Meanwhile, we implement the object localizer without the object correlation graph, the performance is also degraded in both valid unseen and hard valid unseen split. This indicates the correlation mining between target objects and the containers can significantly enhance localization accuracy. Besides, we present the results of Prompter+ with random and Prompter's search policy, which drop significantly in SR and PLWSR. We also provide the results for the settings where the groundtruth location of the interacted objects is given. Our ThinkBot achieves a similar success rate, which indicates the precise prediction of the object location.

Table 4: Comparisons of error modes. Following Min et al. (2022); Inoue & Ohashi (2022), we categorize the failure reasons into 3 main classes to provide further explanation.

| Method | Goal object not found | Interaction failures | Navigation failures |
|---|---|---|---|
| FILM | 53.01 | 6.78 | 19.60 |
| Prompter | 34.39 | 7.72 | 4.62 |
| Prompter+ | 21.10 | 7.56 | 6.95 |
| **ThinkBot** | **18.08** | **7.32** | **6.83** |

**Error Mode:** Table 4 presents the absolute ratio of failure cases caused by different factors in FILM, Prompter, and the proposed ThinkBot. The counterparts' results are taken from their papers Min et al. (2022); Inoue & Ohashi (2022). We categorize failure cases into three types: 'Goal object not found', 'Interaction failures', and 'Navigation failures'. As depicted in Table 4, the occurrence of the 'goal object not found' error substantially decreases (18.08% vs. 21.10%) in our ThinkBot by incorporating the recovered coherent human instruction. Our method does not noticeably impact

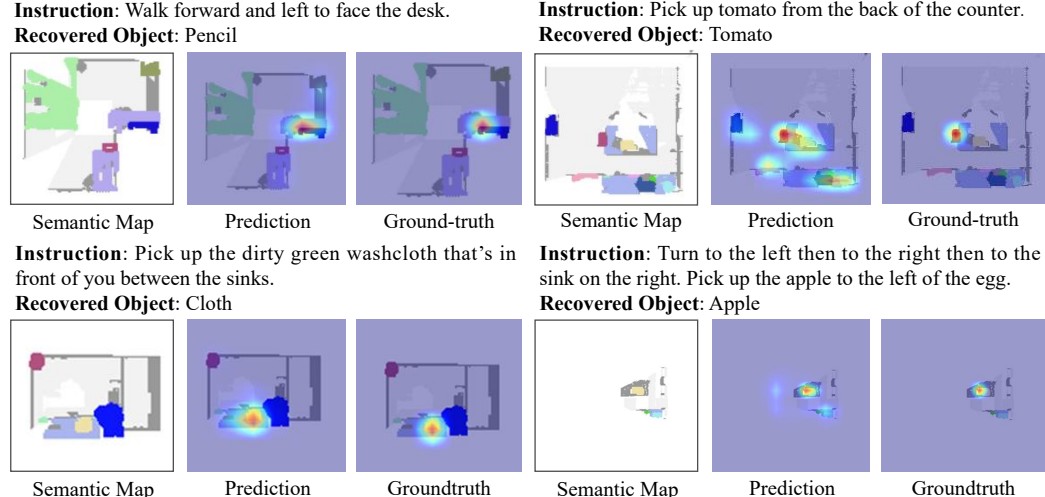

Figure 7: The visualization of the predicted and groundtruth positions of interacted objects, where the partially observed semantic maps are also depicted.

error modes like 'Interaction failures' and 'Navigation failures', since these are unrelated to the instruction-following strategy that this paper focuses on. The results verify the effectiveness of our method in recovering the missing actions to prevent the agent from failure, while maintaining a comparable performance in interaction and navigation to the state-of-the-art methods.

## 4.4 QUALITATIVE ANALYSIS

**Action Sequence Visualization:** We present two qualitative examples of the generated action sequence in Figure 5 from Prompter+ and our ThinkBot. In the left case, the agent is instructed to *'Take a knife. Cut the lettuce in the fridge'*. The results show that the previous agent struggles to complete the task due to the missing 'open' action and interacted object 'fridge'. In contrast, our ThinkBot first reasons the thought chain of human instruction, and then recovers the missing 'open' action and interacted object 'fridge' from the instruction, thus successfully completes the task. In the right case, ThinkBot not only recovers the missing 'open' action and interacted object 'cabinet', but also interacts with the right cabinet that contains the tissue box. The case study demonstrates the effectiveness of ThinkBot in recovering the missing actions and interacted objects from sparse human instruction. In Figure 6, we also showcase an additional complete action sequence of our ThinkBot on the ALFRED valid unseen split. Our ThinkBot can not only recover the missing 'Open' actions but also refine the recovered actions. For instance, our instruction completer outputs 'Close cabinet, go to another cabinet, and open cabinet' when the agent mistakenly opens the wrong cabinet.

**Visualization of the Object Localizer:** Figure 7 shows the groundtruth and the predicted location of the interacted objects from the object localizer, where the object category is generated from the upstream instruction completer. For example in the top case, the interacted object 'pencil' is located on the table, and the object localizer predicts the position of the interacted object with negligible deviation. In cases where multiple instances exist for the recovered object, the object localizer is able to locate all objects in the same category. Meanwhile, our ThinkBot can correctly assign the largest probability to the instance that is described by the instruction, and the human goals can be successfully achieved following the step-by-step human instruction.

## 5 CONCLUSION

In this paper, we have presented a ThinkBot agent that reasons the thought chain for missing instruction recovery in EIF tasks. We design an instruction completer to predict the intermediate actions with interacted objects between incoherent human instruction, and then leverage a multimodal transformer to infer the interaction location for the agent. Extensive experiments in a wide variety of EIF tasks demonstrate the superiority of our method in terms of success rate and execution efficiency.

The main limitations include the low planning frequency during closed-loop thought chain generation, the dependence on the inherent reasoning capabilities of LLMs and the reliance on human instructions instead of active questioning, requiring further alignment for deployment in realistic household scenarios.

ACKNOWLEDGMENTS

This work was supported in part by the National Natural Science Foundation of China under Grant 62125603, Grant 62336004, and Grant 62321005, in part by the Shenzhen Science and Technology Program JCYJ20240813111903006, in part by the Beijing Natural Science Foundation under Grant No. L247009, and in part by the NTU Start up Grant 024303-00001.

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

## A    DETAILS OF THE ALFRED BENCHMARK

In this section, we first provide a concise overview of the input and output space within the ALFRED benchmark, and then analyze the hard valid unseen split we extracted from the valid unseen split.

**Input and Ouput Space.** In a typical ALFRED task, the agent is spawned into a 3D indoor floor plan with a first-person view. To specify a task, the agent receives a high-level goal statement that describes the task's objective and step-by-step instruction that provides detailed explanations. Accompanying these, the agent receives a $300 \times 300$ egocentric RGB frame and outputs actions for each time step. Note that during the test phase, other information such as groundtruth depth images and instance segmentation images is not provided to the agent. The action space of the agent consists of 5 navigation actions, 7 interaction actions, and a STOP action. The navigation actions are MOVEAHEAD, ROTATELEFT, ROTATERIGHT, LOOKUP, and LOOKDOWN. The interaction actions are PICKUPOBJECT, PUTOBJECT, OPENOBJECT, CLOSEOBJECT, TOGGLEOBJECTON, TOGGLEOBJECTOFF, and SLICEOBJECT. If an interaction action is outputted, the agent is required to predict an additional binary object mask for the current RGB frame, so that the agent can interact with the object of the highest IoU score. When the agent takes the STOP action, the simulator will check whether all objects are correctly positioned based on a predefined PDDL domain. The trial will fail if the agent exceeds the 1000-step limit or makes more than 10 errors while attempting to achieve the goal.

**Hard Valid Unseen Split.** As outlined in the main paper, we have created a specific subset named *hard valid unseen* split from the valid unseen split in ALFRED. The hard valid unseen split only includes cases where all target objects are confined within closed containers, to assess the agent's capability to recover missing interactions. These types of cases make up 8.6% in the valid unseen split, 4.8% in the valid seen split, and 5.52% in the train split, indicating the importance for the agent to handle such situations. Consequently, the hard valid unseen split contains 74 trials extracted from the valid unseen split, involving five task types of 'Examine', 'Pick & Place', 'Stack & Place', 'Clean & Place' and 'Heat & Place'. The hard valid unseen split is capable of evaluating the agent's ability to recover missing interactions comprehensively.

## B    DETAILS OF THE PROMPTER+ BASELINE

In the main paper, we have constructed a strong baseline termed Prompter+, and we provide additional implementation details of Prompter+ in this section. Prompter+ is built upon the Prompter codebase, which combines environment-aware memory Kim et al. (2023) and a re-trained object

detector Wang et al. (2023a) with the vanilla Prompter. We utilize the InternImage-XL backbone pretrained on the COCO dataset Lin et al. (2014) with Cascade Mask R-CNN head implemented Wang et al. (2023a) on MMDetection Chen et al. (2019). To collect the dataset for finetuning, we replay the expert trajectories in the ALFRED train split, and record the egocentric image and the groundtruth instance mask at each step. Following Shridhar et al. (2021), we also balance the training samples from each room type in the ALFRED benchmark. For finetuning, we use AdamW optimizer Loshchilov & Hutter (2017) with an initial learning rate of $1 \times 10^{-4}$ and weight decay of $5 \times 10^{-2}$. Please refer to Wang et al. (2023a) for more training details. The whole finetuning process takes one day on 4 NVIDIA 3090 GPUs, where the batch size on each GPU is set to 4.

## C  FULL PROMPT OF THE INSTRUCTION COMPLETER

In this section, we present the full system prompt for the instruction completer in our ThinkBot for reproducibility.

### C.1  COMPONENTS IN THE PROMPT

The system prompt template for our instruction completer consists of the following components:

(1) The role explanation with the emotion prompt Li et al. (2023a);

(2) Definition of the task description:

- High-level goal statement: A string describes the goal of this task;
- Low-level step-by-step instruction: A list contains the whole incoherent human instruction;
- Possible landmarks in this room type: A list contains all possible landmarks in the current room to avoid object hallucination;
- Task Completion: The current task completion progress we provide for the agent to locate the corresponding instruction sentences.

(3) Definition of the agent's current state:

- Global observed landmarks: A dictionary of observed landmarks and their positions, where the uncertainty of estimated object is also given for re-planning;
- Last message: A string contains the failure feedback from the last run, which enables close-loop planning.

(4) Primitive actions: All interaction actions and the related arguments we introduce in Appendix A along with a 'GotoLocation' subgoal for navigation.

(5) Requirements on the response format, where we impose chain-of-thought prompting Wei et al. (2022): We request the large language model to first reason on the current subtask then give a detailed action list. The last predicted subgoal should always be the same as the current subgoal for coherent instruction recovery.

### C.2  FULL PROMPT

The complete system prompt template is shown in Listing 1, and the response format is shown in Listing 2.

Listing 1: Full system prompt for the instruction completer in our ThinkBot. The response format is shown in another listing.

```
You are a helpful household assistant in a game called ALFRED with high
    intelligence. Given human instructions, total subgoals, current
    subgoals, etc., you need to accomplish the current subgoal by giving
    a detailed action list. This is very important to my career.

At each run, you need:
step1. First recognize which target object you need, and determine the
    target location by following the corresponding step-by-step
    instruction sentence or by common sense.
```

```
step2. Give a detailed action list to accomplish the current subgoal, the
     last action should always be the same as the current subgoal.

1. The input contains:
Task description:
- high-level goal statement: describe the goal of this household task.
- low-level step-by-step instructions: describe the step-by-step
     instructions of this household task. The current subgoal is only
     corresponding to one or two sentences in this list, so you do not
     need to use all instructions, just focus on completing the current
     subgoal.
- possible landmarks in this room type: you should only use landmarks in
     this list.
- total subgoals: a list of <subgoal name, arguments>, memorizing all
     subgoals you need to complete.
- completed subgoals: a list of <subgoal name, arguments>, memorizing
     subgoals you have completed.
- current subgoal: <subgoal name, arguments>, the subgoal you are
     currently working on.

My current state:
- observed landmarks: a dict of key <object name> and value (<location>,
     <uncertainty>).
- last message: a string, the message from the last run, success message
     is an empty string.

2. You can ONLY use the following functions. Don't make plans purely
     based on your experience, think about how to use these actions.

GotoLocation(object)
Go to a landmark object.
Augment:
- object: a string, the landmark to go to.

OpenObject(object)
Open an openable object.
Augment:
- object: a string, the receptacle to open. please note that only ['
     Fridge', 'Cabinet', 'Microwave', 'Drawer', 'Safe', 'Box'] are
     openable.

CloseObject(object)
Close an openable object
Augment:
- object: a string, the receptacle to close.

PickupObject(object)
Pick up an object. If the object is inside a closed receptacle, please
     open the receptacle first.
Augment:
- object: a string, the object to pick.

PutObject(object, receptacle)
Put down the holding object to a receptacle.
Augments:
- object: a string, the object to put.
- receptacle: a string, the receptacle to place the object.

SliceObject(object)
Slice a sliceable object with the held knife.
Augments:
- object: a string, the object to slice.

3. Your response should follow the format:
{response_format}
```

```
Ensure that your response can be parsed by Python json.loads

Examples:
...
```

Listing 2: The response format of the instruction completer.

```
{
"thought": "Your thoughts in natural language",
"action_list": [
{"name": "action name", "args": "action arg(s)", "expectation": "describe
    the expected results of this action shortly"},
{"name": "action name", "args": "action arg(s)", "expectation": "describe
    the expected results of this action shortly"}
]
}
```

Table 5: Comparisons with the state-of-the-art methods in success rate on the valid unseen split break down by task type.

| Method | Examine | Pick & Place | Stack & Place | Clean & Place |
|---|---|---|---|---|
| Seq2seq | 0 | 0 | 0 | 0 |
| MOCA | 4.6 | 6.0 | 6.4 | 10.6 |
| HLSM | 36.6 | 34.8 | 4.4 | 11.3 |
| FILM | 29.7 | 16.0 | 2.0 | 33.6 |
| Prompter+ | **80.9** | 46.0 | 32.1 | 71.7 |
| **ThinkBot** (Ours) | 79.2 | **50.0** | **40.4** | **77.9** |
| | Cool & Place | Heat & Place | Pick 2 & Place | **Average** |
| Seq2seq | 0 | 0 | 0 | 0.00 |
| MOCA | 2.8 | 5.1 | 1.2 | 3.78 |
| HLSM | 14.8 | 0.0 | 18.0 | 18.3 |
| FILM | 14.0 | 23.0 | 11.8 | 20.10 |
| Prompter+ | 82.6 | 78.7 | **37.0** | 64.43 |
| **ThinkBot** (Ours) | **88.1** | **86.0** | 33.3 | **67.72** |

# D  ADDITIONAL QUANTITATIVE ANALYSIS

## D.1  PERFORMANCE BY TASK TYPE

We compare the proposed ThinkBot with the state-of-the-art methods on the ALFRED benchmark by task types. The counterparts include end-to-end method Seq2seq Shridhar et al. (2020), MOCA Singh et al. (2021) and modular methods (HLSM Blukis et al. (2022b), FILM Min et al. (2022), Prompter+). The compared results are taken from their original papers Shridhar et al. (2020); Singh et al. (2021); Blukis et al. (2022b); Min et al. (2022). From Table 5, we can observe that our ThinkBot outperforms on almost all task types. For instance, ThinkBot surpasses the state-of-the-art method Prompter+ on five out of seven task types by sizable margins. Especially, ThinkBot succeeds in 40.4% of 'Stack & Place' tasks, which is an absolute improvement of 8.3% compared to the state-of-the-art method Prompter+. While Prompter+ suffers from the sparse human instruction that usually causes execution failure, our ThinkBot reasons the thought chain in the sparse human instruction to recover the missing action descriptions, and successfully complete different tasks. In 'Examine' tasks, the agent is instructed to pick up the target object and toggle on a lamp that is initially off. Since our main focus is not on detecting the status of floor lamps, we adopt a trial-and-error approach by toggling all lamps in the current room following Min et al. (2022); Inoue & Ohashi (2022). This random selection approach introduces variability in the success rate of these tasks. In 'Pick 2 & Place' tasks, the agent is directed to take two instances within the same category and relocate them to a specified location. During the repeated subgoal completion, our language model-based instruction completer may be prone to the hallucination issue, which remains a focus for future improvements.

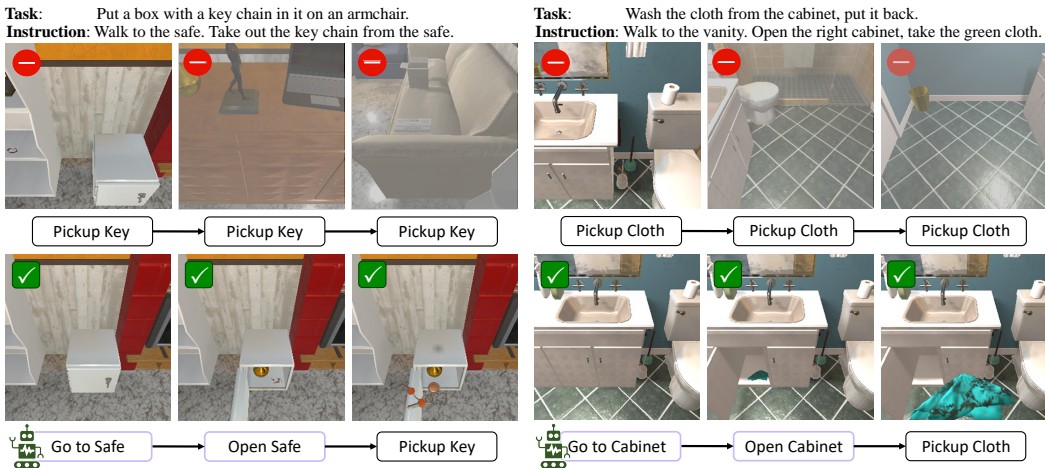

Figure 8: Visualization of the agent action sequences acquired by Prompter+ (top) and our ThinkBot (bottom), where our method can recover the missing actions with interacted instances 'Open Safe' and 'Open Cabinet' to successfully achieve the goal.

## D.2 INFERENCE TIME ANALYSIS

In Table 6, we analyze the time consumption of each component in the whole system. We benchmark the time consumption in a single NVIDIA RTX 4090 GPU. In line with Min et al. (2022), we query the LLM after every 25 steps or subgoal completion to ensure consistency. Hence, the delay of LLM reasoning only affects steps that require important subgoal decisions (slow thinking), while most steps just involve path planning to achieve the subgoal (fast thinking), resulting in an acceptable average time per step. In summary, the whole system executes at 1.37Hz on average.

Table 6: Inference time of different components.

| Module | Segmentation | Depth | Mapping | Path Planning | Object Localizer | Instruction Completer |
|---|---|---|---|---|---|---|
| Avg. Time per Step | 0.20 | 0.0076 | 0.021 | 0.18 | 0.013 | 0.31 |

## D.3 REAL-WORLD APPLICATIONS

The mobile agent consists of a Realman manipulator, a HEXMAN ECHO-Plus mobile base, and a Realsense RGB-D camera for observation. We test sparse human instructions like 'Examine the red block on the table in front of you' to complete one mobile pick & place task. These tasks validate instruction recovery in partially observed scenes via active perception. The primitive skills include: goto(base position), move(arm position) and gripper(openness). An example trajectory is shown in Figure 9, where ThinkBot is able to explore the scene, avoid the obstacle, and complete the mobile manipulation task successfully.

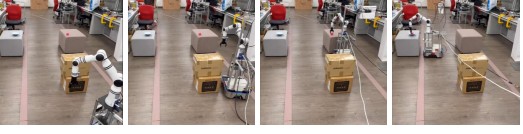

Figure 9: Visualization of the ThinkBot agent performing a mobile pick & place task.

## D.4 IMPACT OF DIFFERENT LLMS

The proposed method is complementary to the internal reasoning abilities of LLMs, which unlocks their spatial reasoning ability by incorporating the uncertainty predicted by the Bayesian object localizer as a feedback signal for closed-loop planning. To mitigate the dependence on the high-capacity base LLM, we implement the instruction completer with a relatively small-

Table 7: Impact of base LLMs.

| Base LLM | Success Rate |
|---|---|
| Llama 3.2 3B | 66.87 |
| GPT-3.5-Turbo | **67.72** |

size and open-sourced LLM Llama 3.2 3B in *valid unseen*. The results on ALFERD's *valid unseen* split are shown in Table 7, where the consistent improvements across different LLMs verify the robustness of ThinkBot.

# E    ADDITIONAL QUALITATIVE ANALYSIS

## E.1    ACTION SEQUENCE VISUALIZATION

We present two more qualitative examples of the generated action sequences from Prompter+ and our ThinkBot in Figure 8. In the left case, the agent is instructed to *'Walk to the safe. Take out the key chain from the safe'*. The results show that the previous agent struggles to complete the task due to the missing 'Open' action and interacted object 'Safe'. On the contrary, our ThinkBot first reasons the thought chain of human instruction, and then recovers the missing 'Open' action and interacted object 'Safe' from the instruction, thus successfully completing the task. In the right case, our ThinkBot not only recovers the missing 'Open' action and interacted object 'Cabinet', but also interacts with the right cabinet instance that contains the green cloth. The case studies demonstrate the effectiveness of ThinkBot in recovering the missing actions and interacted objects from sparse human instruction.

## E.2    A VIDEO FOR COMPLETE TRIAL VISUALIZATION

We provide an additional comprehensive trial visualization in the attached video file (*demo.avi*) selected from the valid unseen split. In the video, the agent is instructed to 'Put a mug with a spoon inside of it on the counter.' However, the mug is stored in the fridge. Our ThinkBot can recover the missing 'Go to Fridge' and 'Open Fridge' subgoal and locate the interacted objects precisely, thus completing the task effectively and efficiently.

