# OpenReview forum: "ThinkBot: Embodied Instruction Following with Thought Chain Reasoning"
_ICLR.cc/2025/Conference — ICLR 2025 Poster_

### Official Review · Reviewer_tHpP · 2024-10-28

**Soundness:** 3
**Presentation:** 3
**Contribution:** 2
**Rating:** 6
**Confidence:** 4

**Summary:**

This paper presents a ThinkBot agent that recovers underspecified instructions through thought chain reasoning and performs embodied instruction following tasks. The proposed method consists of three components: the instruction completer, the object localizer, and the object correlation graph. The authors employ large language models for the instruction completer and a multimodal transformer for the object localizer. Experimental results show that a ThinkBot agent outperforms prior arts on the ALFRED benchmark, and ablation studies demonstrate the effectiveness of each component.

**Strengths:**

- The paper introduces a new modular approach for EIF tasks, a ThinkBot agent. The idea of recovering sparse human instruction is a novel direction.
- The paper conducts extensive analysis, including ablation studies and qualitative analysis, which helps to understand the proposed method.
- The paper is well-written and easy to understand.

**Weaknesses:**

- While the authors demonstrated the effectiveness of the instruction completer in ablation studies (Table 2), the improvements are quite marginal in the valid unseen split. It indicates that recovering sparse human instruction is sometimes helpful, but at the same time, this tells us that it is not a fundamental problem in EIF tasks.

**Questions:**

- What makes the difference between ThinkBot and LLM-Planner in high-level planning (Table 3)?

---

> ### Author Response · Authors · 2024-11-25
> **Author Response**
>
> Thanks for your insightful review! We address your concerns as follows.
>
> **Q1. The improvements are marginal in valid unseen.**
>
>
> The ablated results of the proposed techniques indicate a significant drop from 67.72% to 64.92% (-2.80%) in success rate, which is a great performance gap observed in ALFRED.
> As a reference, the key idea of the semantic search policy in FILM paper [1] reports a smaller performance gain of +0.24% (from 19.85% to 20.09%).
> Besides, the ablated results of the instruction completer are significant in *hard valid unseen* split with an absolute degradation of 22.97%, which is actually more realistic as target objects are often tidied away and stored in receptacles in daily household tasks.
> Moreover, the problem of understanding sparse human instruction is essential in diverse multi-model embodied tasks including embodied instruction following, high-level planning, object navigation and even robotic manipulation. We have conducted extensive experiments in these domains to verify the generalizability of the proposed method (The results are shown in the responses to Reviewer oWwC-Q6 and Review pZ3K-Q2).
>
>
> **Q2. Difference between ThinkBot and LLM-Planner in high-level planning.**
>
> In terms of high-level planning accuracy, the difference between Thinkbot and LLM-Planner is mainly caused by two reasons:
> - LLM-Planner directly mimics the retrieved few-shot examples. Besides, Thinkbot generates explicit thought chains to recover the sparse human instruction before predicting the next subgoal, which enhances high-level planning with structured reasoning.
>
> - LLM-Planner only leverages a discrete observed object list, ignoring other information. On the contrary, Based on the explicit thought chain, Thinkbot processes the uncertainty information of the observed objects to refine the thought chain, so that the predicted subgoal can lead to efficient task completion.
>
> For example:
> ```
> Instruction: Examine a box of tissues using a floor lamp.
> Thinkbot: Pickup TissueBox (Attempt) -> Open Cabinet -> Pickup TissueBox -> ToggleOn Lamp (HLPACC=100%)
> LLM-Planner: Pickup TissueBox -> ToggleOn Lamp (HLPACC=0%)
> Groundtruth: Open Cabinet -> Pickup TissueBox -> ToggleOn Lamp
> ```
> LLM-Planner fails when the few-shot examples do not contain the case of a TissueBox stored in a cabinet. In contrast, Thinkbot can infer the intermediate 'Open Cabinet' by reasoning about the observed objects (Cabinet, Sofa, Safe) and the localizer's output (Cabinet with high uncertainty).
>
> [1] FILM: Following instructions in language with modular methods, ICLR 2022.

---

> ### Author Response · Authors · 2024-11-29
> **Author Follow-up**
>
> Dear Reviewer,
>
> We would like to ask if your concerns regarding the ablation study and high-level planning addressed, and if there is anything preventing you from increasing your score.
> Please let us know, and thank you for your time!

---

> > ### Comment · Reviewer_tHpP · 2024-12-03
> > **Thanks for the response!**
> >
> > I would like to thank the authors for the response of my concern and question. Currently, I do not have a follow-up comments or questions about the rebuttal.

---

> > > ### Author Response · Authors · 2024-12-03
> > > **Thanks!**
> > >
> > > Thanks for your positive reply! We truly appreciate your thoughtful and constructive review.

---

### Official Review · Reviewer_oWwC · 2024-11-01

**Soundness:** 3
**Presentation:** 3
**Contribution:** 2
**Rating:** 6
**Confidence:** 4

**Summary:**

The paper presents ThinkBot, a system designed to infer missing actions from incomplete human instructions using LLMs and to predict target object locations for these inferred actions. ThinkBot employs a structured CoT prompt to obtain the missing actions using LLMs. With the actions and target objects identified, the multimodal object localizer creates a positional heatmap the given instruction, and the inferred action. ThinkBot achieves improvements over previous methods in the ALFRED benchmark.

**Strengths:**

- ThinkBot tries to figure out missing actions from instructions, which is an important and challenging problem of embodied instruction following.
- ThinkBot shows a strong improvement over the previous methods (yet the baseline, Prompter+, on which ThinkBot is built already beats SoTA, see weaknesses).
- The paper is well structured and shows clear presentation.

**Weaknesses:**

- It looks like the instruction completer finds where is a target object by addressing missing information in the goal statement, but the step-by-step instructions usually contain the missing information. Why not just directly parse the missing information in the step-by-step instructions? Why do we still need the proposed instruction completer?

- Previous work (Prompter) also similarly finds the next receptacle by using language models (BERT) by guessing where a target object is likely to be. Instead, the instruction completer uses LLM with CoT. We may use better language models with prompting techniques for better results, but this is expected.

- Why not just using semantic information in the spatial semantic map, not predicting where is a target by the multimodal object localizer?

- The ablated performance of the multimodal object localizer and the correlation graph seems marginal in valid unseen but not in Hard Valid Unseen. It seems like the model is designed for closed-receptacle cases. And why are they marginal?

- I think that the authors can plug the proposed module in the other baselines like FILM and CPEM. Improvements over these models may indicate good generalizability of the proposed approach.

- The proposed model is evaluated in a single benchmark. I'm not sure if this approach works well in other tasks.

**Questions:**

See weaknesses above.

---

> ### Author Response · Authors · 2024-11-25
> **Author Response**
>
> Thanks for your valuable feedback! We provide answers to your questions below.
>
> **Q1. Why not use step-by-step instructions directly.**
>
> The step-by-step instructions still does not align with the subgoal sequence and often misses some information like the target instance position.
> For example, 'Grab an egg out of the fridge.' ignores 'Close Fridge'.
> Moreover, our method supports goal statements only without low-level instructions, which is a more convenient and realistic way in household scenarios. To verify this, we conduct comparisons on different methods:
> | Method | Low. Inst. + High. Goal | High. Goal Only |
> | :--- | :---: | :---: |
> | CPEM | 46.11 | 43.69 |
> | Prompter | 45.32 | 41.53 |
> | Thinkbot | 67.72 | 67.11 |
>
> Thinkbot achieves the minimal performance gap when removing the low-level instructions, showing its superior reasoning ability given extremely sparse human instructions.
>
>
> **Q2. Better language models with prompting techniques for better results is expected.**
>
> Prompter uses BERT to identify the goal object, while utilizing GPT-Neo as the large language model to guess which receptacle a target object is likely to be in.
> We replace GPT-Neo with advanced GPT-4o in Prompter, and use advanced prompting techniques:
> | Method | SR |
> | :--- | :---: |
> | Prompter+ | 64.43 |
> | Prompter+ w/ GPT-4o | 64.92 |
> | Thinkbot | 67.72 |
>
> By incorporating the advanced language model and prompting techniques, Prompter improves from 64.43% to 64.92%, which is neglectable as the cooccurence probablities are basically the same as the vanilla version.
> Besides, the proposed framework complements the language model, please also refer to the response to Reviewer pZ3K-Q1, where we implement Thinkbot with different LLMs and observe consistent improvements.
>
>
> **Q3. Why not just use information in the semantic map.**
>
> The semantic map is often partially-observed, and the instance that lies in closed receptacle is not shown in the built semantic map.
> Therefore, existing methods [1,2,3] study different localizers to infer the possible position of the target.
> However, these methods perform planning and localization separately, which struggles with error interactions due to the open-loop nature.
> To this end, we introduce a graph-based multimodal object localizer based on the partially-observed semantic map and the rationales from the instruction completer, and outputs the Bayesian uncertainty for the instruction completer to conduct closed-loop planning.
>
> **Q4. Marginal ablated performance in valid unseen.**
>
> The ablation of the object localizer results in a noticeable performance drop in success rate (-0.73%). For comparison, the FILM paper [1] reports a smaller performance gain of +0.24% (from 19.85% to 20.09%) by incorporating it semantic search policy.
> The object localizer is designed to infer the positions of interacted objects in a partially-observed map, enabling efficient exploration. In long-horizon tasks of ALFRED, the environment gradually becomes close to fully observed as the task progresses. Nevertheless, the performance gaps are more pronounced in PLWGC (-0.87%) and PLWSR (-0.86%).
> Moreover, the ablated results of the object localizer are significant in *hard valid unseen* split with a relative degradation of 29.39%, which is actually more realistic as objects are often tidied away and stored in receptacles in daily household tasks.
>
>
> **Q5. Implement the proposed module in other baselines.**
>
> We incorporate Thinkbot with FILM and CPEM, and report the success rates in the leaderboard split *test unseen* in ALFRED:
> | Method | SR |
> | :--- | :---: |
> | FILM | 27.80 |
> | FILM w/ Thinkbot | 34.66 |
> | CPEM | 46.11 |
> | CPEM w/ Thinkbot | 51.01 |
>
> Note that here we use the non-templated version of CPEM for re-planning.
> The consistent improvements across FILM, CPEM and Prompter indicate the generalizability of the proposed approach.
>
> **Q6. Single benchmark.**
>
> We have tested Thinkbot in ALFRED, ActioNet, and a real-world robotic manipulation task (please check pZ3K-Q2), and demonstrates significant improvements across various domains.
> We further validate it in a subset of HM3D ObjectNav benchmark [4], where we use 80 scenes for training and evaluate 20 scenes in the valid split:
> | Method | SR |
> | :--- | :---: |
> | SemExp | 37.9 |
> | SemExp w/ Thinkbot | 42.6 |
>
> By incorporating Thinkbot with the well-recognized baseline SemExp [5], we boost the performance by a large margin of $4.7$\%, which indicates the generalizability of the proposed method.
>
> [1] FILM: Following instructions in language with modular methods, ICLR 2022.
>
> [2] Prompter: Utilizing Large Language Model Prompting for a Data
> Efficient Embodied Instruction Following, arXiv 2022.
>
> [3] Following Natural Language Instructions for Household Tasks With Landmark Guided Search and Reinforced Pose Adjustment, RA-L 2022.
>
> [4] Habitat-matterport 3d semantics dataset, CVPR 2023.
>
> [5] Object Goal Navigation using Goal-Oriented Semantic Exploration, NeurIPS 2020.

---

> ### Author Response · Authors · 2024-11-29
> **Author Follow-up**
>
> Dear Reviewer,
>
> We would like to ask if your concerns regarding the experimental setting and the generalizability of the proposed method addressed, and if there is anything preventing you from increasing your score.
> Please let us know, and thank you for your time!

---

### Official Review · Reviewer_ZBDZ · 2024-11-03

**Soundness:** 3
**Presentation:** 3
**Contribution:** 2
**Rating:** 5
**Confidence:** 2

**Summary:**

This paper proposes a novel agent for Embodied Instruction Following (EIF) tasks. The method uses thought chain reasoning to complete missing action descriptions and employs a multimodal object localizer for enhanced object interaction. Extensive experiments demonstrate better performance in both success rate and execution efficiency compared to previous methods.

**Strengths:**

The proposed pipeline is technically sound. Using LLM to understand free-form instructions is an effective way to reason about actions in unstructured environments. And the proposed object localizer complements the LLM by addressing its spatial understanding limitations.

**Weaknesses:**

1. While effective, this work appears to be a combination of LLM with a localizer, which might lack of novelty.

Existing multimodal LLMs (MLLMs) can achieve better reasoning performance than the LLM used here. The authors employed GPT-3.5, but using a state-of-the-art MLLM like GPT-4o could improve performance.

**Questions:**

What if you take input the instruction with current egocentric image to gpt-4o? I believe it might improve the overall reasoning performance.

---

> ### Author Response · Authors · 2024-11-25
> **Author Response**
>
> Thanks for your review! Here, we respond to your comments.
>
> **Q1. Lack of novelty.**
>
> We respectfully argue that our work introduces novelty in addressing the challenges of enhancing LLMs with spatial reasoning capabilities for interactive agents, which is non-trivial.
> Naive combination of LLM with a localizer struggles to recover attainable subgoals in EIF, because it requires:
> - Spatial Reasoning in Partially-observed Scene: Sparse human instructions often omit details of exploration in partially-observed environments. This requires the agent to infer missing information based on spatial clues, which is non-trivial for LLMs due to the lack of explicit guidance in both language and vision.
> - Executability for low-level APIs: The embodied instruction following agents often utilize a low-level API (e.g., a pretrained detector) for interaction, which can fail in unstructured scenes (e.g., invalid mask from a weird direction), requiring precise uncertainty feedback to prevent execution errors.
>
> To tackle these, we propose an instruction completer that guides the LLMs to reason intermediate subgoals to reduce uncertainty of the target in partially-observed scene, and introduce a Bayesian object localizer to estimate the uncertainty feedback precisely.
> To conclude, the proposed reasoning and feedback framework complements the internal capabilities of LLMs (please refer to pZ3K-Q1), and shows significant improvements in a wide range of embodied tasks, including ALFRED, ActioNet, Habitat and real-world robotic manipulation (please check oWwC-Q6 and pZ3K-Q2).
>
> **Q2. Try GPT-4o.**
>
> We input the instruction with current egocentric image to the most advanced language model GPT-4o:
> | Method | SR |
> | :--- | :---: |
> | GPT-4o w/ ego. obs. | 24.00 |
> | GPT-4o w/ sem. map | 32.40 |
> | Thinkbot | 67.72 |
> | Thinkbot w/ GPT-4o | 68.94 |
>
> Due to the high cost of planning each low-level action step with GPT-4o, we evaluate the compared methods in the main *valid unseen* split for comparisons.
> To avoid hallucination, we let GPT-4o choose a low-level action from 'MoveForward', 'RotateLeft' and 'RotateRight' for navigation, and select a target object name from the detected object list for interaction.
> We also provide the distance to each object in the viewing frustum from depth estimation, which can avoid interacting with a remote object.
> We sample $k=9$ successful demonstrations using $k$-NN from training set for in-context learning, where the pairwise similarity is estimated by a frozen BERT as per [1].
> We further simplify the problem by letting MLLMs reason on a dilated online semantic map, addressing the issues of collision and spinning around due to the lack of proprioception and memory.
> However, both trials of directly using GPT-4o lead to relatively low performances, where the main error mode comes from the internal reasoning of GPT-4o can not achieve efficient exploration.
> On the contrary, Thinkbot leverages spatial uncertainty feedback for closed-loop planning, which is complementary to the internal reasoning ability of LLMs (MLLMs).
> As a result, by incorporating more cutting-edge LLMs, our method can consistently improve its performance, demonstrating the versatility of the proposed reasoning and feedback framework.
>
> [1] LLM-Planner: Few-Shot Grounded Planning for Embodied Agents with Large Language Models, ICCV 2023.

---

> ### Author Response · Authors · 2024-11-29
> **Author Follow-up**
>
> Dear Reviewer,
>
> We would like to ask if your concerns regarding more comparisons with advanced MLLMs addressed, and if there is anything preventing you from increasing your score.
> Please let us know, and thank you for your time!

---

### Official Review · Reviewer_pZ3K · 2024-11-03

**Soundness:** 3
**Presentation:** 3
**Contribution:** 2
**Rating:** 6
**Confidence:** 4

**Summary:**

The paper introduces ThinkBot, an agent designed for Embodied Instruction Following (EIF) that effectively executes human instructions in complex environments by reconstructing missing action descriptions through the chain of thought. The authors introduce an instruction completer using large language models to predict missing actions and an object localizer to determine object positions for agent interaction. Extensive experiments show that ThinkBot surpasses existing state-of-the-art EIF methods in both success rate and execution efficiency. Key contributions include the development of the ThinkBot agent, the instruction completer for coherent instruction generation, and the object localizer for precise spatial guidance, all enhancing success rates in EIF tasks.

**Strengths:**

- The paper presents a novel approach to Embodied Instruction Following (EIF) by introducing the concept of a "thought chain" to recover missing action descriptions from sparse human instructions. Integrating an instruction completer and an object localizer enhances the agent's ability to follow instructions in complex environments. This is an effective way to address the coherence issue in action plans.
- The experiments in the paper demonstrate the effectiveness of ThinkBot through extensive testing on the ALFRED benchmark, showing significant improvements in success rates and execution efficiency over state-of-the-art methods.
- The paper is well-structured. The figures and diagrams, such as the pipeline of ThinkBot and the comparison with conventional methods, are clear and effectively illustrate the proposed methods and results. The writing is coherent, and the instructions for reproducibility, including the full system prompt for the instruction completer, contribute to the transparency and clarity of the research.

**Weaknesses:**

- The introduced approach heavily relies on the inherent reasoning capabilities of LLMs. If the LLMs are not trained on diverse or comprehensive datasets, it could lead to biases or limitations in the agent's performance. The authors need to demonstrate the robustness of the prompt to different LLMs, e.g., Llama 3, and other open-sourced LLMs (gemma2, qwen2.5).
- While the paper demonstrates strong performance in simulated environments, there might be concerns about how well ThinkBot's approach translates to real-world applications, where the complexity and variability can be significantly higher. Besides, the time cost of the reasoning process should be considered for real-world applications. However, LLM-based reasoning will lead to a high delay in the inference time. The use of LLMs and complex reasoning processes may require significant computational resources, which could limit the scalability of the ThinkBot system, especially in resource-constrained embodied systems.
- The paper does not extensively discuss how ThinkBot handles errors or recovers from incorrect actions once taken. This is an important aspect of robust autonomous agent design. It is important to make the agent self-improve during the interactions, instead of manually adjusting the prompt. This is also important to generalize the reasoning process to other scenarios or tasks. It is necessary to conduct experiments demonstrating the agent's ability to learn from mistakes (adding noise to actions) and improve performance over time without manual prompt adjustments or discuss how the reasoning process could be made more generalizable.

**Questions:**

- How does the performance of ThinkBot vary when using different LLMs with potentially different training datasets? Have you tested the robustness of the prompt with LLMs such as Llama 3 or other open-sourced LLMs?
- How does the ThinkBot approach handle the increased complexity and variability of real-world environments compared to simulated ones?
- What is the inference time for the LLM-based reasoning process, and how does this delay impact the feasibility of ThinkBot for real-world applications? How does the computational resource requirement of the ThinkBot system impact its scalability, especially when considering the constraints of embodied systems in real-world scenarios?
- Can the agent learn from its mistakes to improve its performance over time, and is there a mechanism for self-improvement during interactions?
- How to address the generalization of the reasoning process to new scenarios or tasks, ensuring that the agent can adapt and learn from a wide range of situations?

---

> ### Author Response · Authors · 2024-11-25
> **Author Response**
>
> Thank you for the detailed and constructive review! We answer your comments below.
>
> **Q1. Different LLMs.**
>
> The proposed method complements the internal abilities of LLMs, which unlocks their spatial reasoning ability by incorporating the uncertainty predicted by the Bayesian object localizer as a feedback signal for closed-loop planning.
> We implement the instruction completer with a relatively small-size and open-sourced LLM Llama 3.2 3B in *valid unseen*:
> | Base LLM | SR |
> | :--- | :---: |
> | Prompter+ | 64.43 |
> | GPT-3.5-Turbo | 67.72 |
> | Llama 3.2 3B | 66.87 |
> The consistent improvements across different LLMs verify the robustness of Thinkbot.
>
> **Q2. Real-world applications.**
>
> The simulation benchmarks ALFRED and ActioNet used in our paper require a mobile manipulation agent, which indeed pose challenges for deployment.
> But Thinkbot is a general reasoning framework that can be easily extended to other environments with atomic skills.
> To verify its practicality, we test it in a constrained yet realistic language-conditioned robotic manipulation task *pick up the blue cube*.
> The atomic skills used are move(position) and gripper(openness).
> For simplicity, the object localizer is implemented using a pretrained Grounded-SAM [1], with its confidence score serving as the uncertainty measure.
> In each test episode, the object locations are randomized, and distractors (e.g., blocks of other colors) are added to evaluate the generalizability:
> | Method | SR |
> | :--- | :---: |
> | Progprompt [2] | 2/5 |
> | Progprompt w/ Thinkbot | 4/5 |
>
> Thinkbot enhances programmatic generation by recovering dense and actionable instructions like slightly adjusting the gripper pose for better grasping based on the visual feedback, leading to a sizable improvement.
> We have updated the real-world execution videos in the supplementary material.
>
> **Q3. Inference time.**
>
> We benchmark the time consumption in a single NVIDIA RTX 4090 GPU.
> In line with [3], we query the LLM after every 25 steps or subgoal completion to ensure consistency.
> Hence, the delay of LLM reasoning only affects steps that require important subgoal decisions (slow thinking), while most steps just involve path planning to achieve the subgoal (fast thinking), resulting in an acceptable average time per step:
> | Base LLM | Avg. Time per Step |
> | :--- | :---: |
> | GPT-3.5-Turbo | 0.31 |
> | Llama 3.2 3B | 0.17 |
>
> We also provide the time analysis of each component in the whole system:
> | Component | Avg. Time per Step |
> | :--- | :---: |
> | Segmentation | 0.20 |
> | Depth | 0.0076 |
> | Mapping | 0.021 |
> | Path Planning | 0.18 |
> | Object Localizer | 0.013 |
> | Instruction Completer | 0.31 |
> | Total | 0.7316 |
>
> In summary, the whole system executes at 1.37Hz on average.
> The INT4 version of Llama 3.2 3B only requires 1.75 GPU memory, which can be easily deployed in consumer-size GPUs.
>
> **Q4. Error recovery and self-improvement.**
>
> Figure 6 in the manuscript shows when the agent mistakenly opens the wrong cabinet instance, our instruction completer outputs 'close cabinet, go to another cabinet, and open cabinet' to recover from the error interactions.
> To validate the error recovery of Thinkbot, we add noise to the low-level actions: with 20% probability, the agent randomly chooses 'RotateRight' or 'RotateLeft'.
> 'MoveAhead' and interaction actions are not randomized, as their errors count and don't change the environment:
> | Method | SR |
> | :--- | :---: |
> | Thinkbot | 67.72 |
> | Thinkbot w/ rand. actions | 65.04 |
>
> The slight performance drop shows the robustness of the proposed agent.
>
> In terms of self-improvement, the current method we evaluate on ALFRED and ActioNet is static and re-intialized at the beginning of each episode.
> However, we can easily combine the proposed method with Voyager [4] by storing successful subgoal trajectories in a ever-growing knowledge base for few-shot learning.
> Besides, the densified instruction can also be used to finetune the instruction completer, which formulates a self-improvement loop like STaR [5].
>
> **Q5. Generalization of the Reasoning Process.**
>
> Our prompts specify the general reasoning and feedback format, and there is no information leakage or manual adjustment.
> To address the generalization of the reasoning process to new scenarios or tasks, we can just replace the atomic skills in the prompt, then the agent can complete the tasks autonomously.
> The superior performances across diverse test suites including ALFRED, ActioNet, Habitat (view oWwC-Q6) and real-world robotic manipulation verify the generalizability of the proposed method.
>
> [1] Grounded SAM: Assembling Open-World Models for Diverse Visual Tasks, arXiv 2024.
>
> [2] ProgPrompt: Generating Situated Robot Task Plans using Large Language Models, ICRA 2023.
>
> [3] FILM: Following instructions in language with modular methods, ICLR 2022.
>
> [4] Voyager: An Open-Ended Embodied Agent with Large Language Models, TMLR 2023.
>
> [5] STaR: Bootstrapping Reasoning With Reasoning, NeurIPS 2022.

---

> > ### Comment · Reviewer_pZ3K · 2024-12-01
> >
> > Thanks for your detailed response. I still have some concerns about the real-world deployment. To be specific, the experiments on the simulator are mainly about a mobile robot navigating in an indoor room. However, the real-world demo is about robotic arm control. The mobile robot setting requires the agent to handle partial observation with an active perception policy. However, the robotic arm setting only needs to parse the state from the image from a static view. Therefore, the difficulty of these two task settings is also quite different. Besides, the whole system only executes at 1.37Hz on average,  which will not satisfy the requirement of controlling the robot in real time, particularly for the mobile robot. Besides, more diversity in real-world testing environments is required. For example, the robotic arm should be evaluated using various instructions.  So the real-world demonstration of a robotic arm is not convincing enough to demonstrate the generalization of the overall framework in real-world scenarios.
> >
> > Moreover, as for "we query the LLM after every 25 steps", how do you get the 25 steps is the best choice? What will happen if you use higher or lower numbers of steps? Should we tune this hyperparameter when transferring to new tasks or environments?

---

> ### Author Response · Authors · 2024-11-29
> **Author Follow-up**
>
> Dear Reviewer,
>
> We would like to ask if your concerns regarding more generalization studies addressed, and if there is anything preventing you from increasing your score.
> Please let us know, and thank you for your time!

---

### Public Comment · ~Zhiwei_Jia1 · 2024-11-18
**An interesting approach for embodied instruction following**

Hi authors,

Thanks for the great work! Please consider citing a relevant work [1] that takes an approach similar to the proposed multimodal object localizer by taking egocentric visual features and instance segmentation results to produce 2D semantic maps for downstream tasks in ALFRED.

[1] Learning to Act with Affordance-Aware Multimodal Neural SLAM, IROS 2022

---

> ### Author Response · Authors · 2024-11-25
> **Thanks for your Interest, we have added the citation**
>
> Thanks for your interest in our work. We sincerely appreciate your pioneering work in incorporating affordance-aware semantic map into the field of embodied instruction following, and include it in the revision (L101 in page 2). We are welcome to further discussion!

---

### Author Response · Authors · 2024-11-26
**Author Response Summary**

We express our gratitude to the four reviewers for their valuable feedback and recognition of our strengths, including 'showing significant improvements in success rates and execution efficiency over state-of-the-art methods' (Reviewer pZ3K), 'the proposed pipeline is technically sound' (Reviewer ZBDZ), 'an important and challenging problem of embodied instruction following' (Reviewer oWwC), and 'the paper is well-written and easy to understand' (Reviewer tHpP).

We conduct and report additional experiments in the manuscript, and briefly conclude the major results here:
- In response to Reviwer pZ3K's question about inference speed, we report the average time per step of different LLMs and other components in the pipeline. The system executes at 1.37Hz on average, which is feasible in real-world settings.
- In response to Reviwer pZ3K and oWwC's question about generalizability, we implement Thinkbot in HM3D ObjectNav benchmark and a real-world language-conditioned manipulation task. We observe that Thinkbot enhances representative baselines with consistent improvements.
- In response to Reviewr pZ3K, ZBDZ and oWwC's question about the usage of different LLMs, we experiment with open-sourced LLMs, and include baselines using MLLMs and Prompter+ with advanced LLMs. The results confirm the proposed reasoning and feedback framework is robust and independent of specific LLM choices.

The answers are as follows. Please let us know if you have any additional questions.

---

### Meta-Review · Area_Chair_Y7ox · 2024-12-23

**Metareview:**

The paper proposes ThinkBot, a framework for embodied instruction following (EIF) that uses chain-of-thought-like reasoning to follow instructions in which some of the actions may not be explicitly conveyed. The "instruction completer" uses a large language model (LLM) to infer missing actions and object localizer helps to identify the location of the target objects. In experiments using the Alfred simulator, ThinkBot outperforms EIF baselines in terms of success rate and task efficiency.

The paper is highly topical---the use of large language models to facilitate instruction-following in embodied settings continues to attract significant attention. ThinkBot differs from some of this work in its focus on explicitly inferring actions that are important to the success of a task, but may not be mentioned explicitly, something that regularly happens in practice. The reviewers acknowledged the importance of this problem and at least two reviewers appreciated the contributions of the thought chain approach to recovering latent action descriptions. The reviewers noted the paper's extensive experimental evaluation, which demonstrates the effectiveness of ThinkBot relative to contemporary baselines. Meanwhile, at least two reviewers commented that the paper is well structured and easy to read. However, the reviewers raised some concerns about the paper as originally submitted. Among them is the method's sensitivity to the choice of LLMs and whether using an improved LLM would match the gains provided by ThinkBot. During the rebuttal, the authors provided experimental results with different language models, including some with GPT-4o, which suggest that ThinkBot's performance gains are robust, at least for now. Another concern was whether the method could generalize to real-world settings, due to their higher variability and complexity, as well as be capable of inference rates that are sufficiently fast for real-time use. The authors made an effort to address these concerns, included by reporting new experimental results, which the reviewer found satisfying. However, the AC disagrees with the claim that an operating frequency of 1.5 Hz is sufficient for practical use. Additionally, there were questions about how ThinkBot recovers after executing incorrect actions, which were addressed with results that involve deploying ThinkBot with an agent whose actions exhibit some randomness and, in turn, are more prone to error.

Overall, the paper addresses an important problem in embodied instruction following. The method itself is interesting, though the extent to which it is novel is not entirely clear. That said, the paper reveals respectable performance gains over existing methods. Most of the reviewers primary concerns were addressed during the rebuttal period.

The AC acknowledges that Reviewer ZBDZ's criticisms were not fully substantiated and that they did not respond to the authors' rebuttal or their attempts to engage the reviewer in discussion. As a result, the AC placed less weight on their review when making the recommendation.

**Additional Comments On Reviewer Discussion:**

With the exception of Reviewer ZBDZ, all of the reviewers responded to the authors' rebuttal. As noted in the meta review, the AC placed less weight on Reviewer ZBDZ's comments because they were not well substantiated and because they did not take part in the discussion.

---

### Decision · Program_Chairs · 2025-01-22

Accept (Poster)